# Implementation Risk Factors of Collaborative Housing in Poland: The Case of 'Nowe Żerniki' in Wrocław

**Piotr Lis** [1] [iD]**, Zuzanna Rataj** [2,*] [iD] **and Katarzyna Suszyńska** [3] [iD]

1   Department of Business Activity and Economic Policy, Poznan University of Economics and Business, 61-875 Poznan, Poland; piotr.lis@ue.poznan.pl
2   Department of Sociology and Business Ethics, Faculty of Economics, Poznan University of Economics and Business, 61-875 Poznan, Poland
3   Department of Investment and Real Estate, Faculty of Management, Poznan University of Economics and Business, 61-875 Poznan, Poland; katarzyna.suszynska@ue.poznan.pl
*   Correspondence: zuzanna.rataj@ue.poznan.pl; Tel.: +48-61-856-9000

**Abstract:** The main aim of this study is to elucidate implementation risk factors of collaborative housing in Poland. The research is based on Rogers' framework supported by Kim and Mauborgne's utility map. On this basis, in-depth interviews were carried out with the leader of a housing project called 'Nowe Żerniki' in Wrocław, residents of collaborative housing, third sector actors, architects and researchers. An analysis of potential demand for collaborative housing was conducted using focus groups for a selected group of students who will make housing choices in the coming years. In their work, the authors assess the factors inhibiting the rate of diffusion of collaborative housing in Poland, point out the utility constraints and try to formulate the basic conditions for socio-technical transitions in the area of grassroots housing. The results indicate that collaborative housing in Poland requires a friendly and supporting environment, credit institutions with experience in handling innovative housing projects and experienced leaders. The connection to the place and the local capital are crucial factors for all the key actors of the collaborative housing process.

**Keywords:** collaborative housing; diffusion of innovations; utility maps; Poland

## 1. Introduction

In Poland, an average of 132,000 dwellings per year were built in the years 1991–2021 (GUS 2022). An increase in the number of dwellings built in 2016–2021 was a record in Poland since the political changes in the early 1990s (Lis et al. 2021). In those five years, almost one million dwellings were constructed, constituting over six per cent of the national housing stock and an equivalent of the entire housing stock in Warsaw (Lis 2021). It shall be highlighted that both the existing housing stock in Poland and the structure of new housing developments are entirely geared towards owner-occupied housing. The investment boom was accompanied by a dynamic increase in housing prices nationwide, reducing affordability for increasingly broad social groups, especially young people, families with two or more children and the elderly (Lis et al. 2020). Public housing, which in Poland is based on the rental of dwellings, accounted for only two per cent of dwellings built over the whole period (GUS 2021). The role of housing cooperatives in Poland, which have a long tradition of activity since 1890, was reduced to the administration of the stock (Coudroy de Lille 2015).

Against this background, different initiatives are emerging in Poland to tackle the housing problems of the people who are not satisfied with the current housing options available on the market and facilitate a transition to sustainability in the area of housing. The most advanced Polish example is the collaborative housing in Wrocław named 'Kooperatywa Mieszkaniowa Nowe Żerniki'. It is a part of a model housing estate 'Nowe Żerniki' initiated in 2011 by a group of 40 architectural bureaus with active participation

of the city of Wrocław, following the tradition of the 1929 WuWa exhibition 'Wohnungs-und Werkraumausstellung' (Housing and Workplaces). 'Nowe Żerniki', co-created from the beginning by future residents, has become a place for young people and for seniors, with a market, nursery, senior citizens' house and a canteen for the elderly. In just a few years, an engaged civil society on a micro-scale was created. It seems that the model practices developed in Wrocław, with multiple architectural awards, should have spread to other housing projects in other parts of Poland. Unfortunately, it has not happened. This phenomenon is part of a much broader context than the Polish one. The question arises about the factors that have inhibited the spread of this type of an innovative project on a wider scale. Moreover, the question arises as to how niches emerge, persist and move into the socioeconomic mainstream? It has been proven that socio-technical transitions entail changes in markets, user practices, policy and cultural meanings beyond the adaptation of new technology (Geels 2004). Nevertheless, the existing systems are stabilised by lock-in mechanisms that relate to sunk investments, behavioural patterns, vested interests, infrastructure, favourable subsidies and regulations (Unruh 2000). Transitions to sustainability do not come easily because the existing systems are characterised by path dependence and are oriented towards incremental innovation along predictable trajectories (Rayner and Malone 1998).

According to (Rogers 1962, 2003), transitions to sustainability can be supported by five factors determining the rate of diffusion of innovation, such as: observability, complexity, compatibility, trialability and relative advantage. Observability is the degree to which the results of an innovation are visible to others, whereas complexity is the degree to which an innovation is perceived as difficult to understand and use (Rogers 2003). These attributes are related to the public awareness of collaborative housing. Compatibility is the degree to which an innovation is perceived as being consistent with the existing values, past experiences and needs of potential adopters, whereas trialability is the degree to which an innovation may be experimented with on a limited basis (Rogers 2003). These attributes are related to the ability to use and adapt collaborative housing. Furthermore, the relative advantage is the degree to which an innovation is perceived as better than the idea it supersedes (Rogers 2003). This is closely related to the assessment of the cost-effectiveness of collaborative housing in relation to other forms of satisfying housing needs. A utility map created by (Kim and Mauborgne 2004, 2017) is used to identify pain points and utility spaces for innovation. The attempt to apply a utility map supporting Roger's framework to collaborative housing has not been found in the literature so far.

The main aim of this study is to elucidate limitations to the diffusion of collaborative housing in Poland. By focusing on the case of the 'Kooperatywa Mieszkaniowa Nowe Żerniki' in Wrocław, the authors assess the factors inhibiting the rate of diffusion of collaborative housing, point out the utility constraints of collaborative housing and try to formulate the basic conditions for socio-technical transitions in the housing sphere in Poland.

Collaborative housing, also known as grassroots housing or participatory housing, is a special kind of the social housing sector, other than public housing, which meets housing needs on an institutional basis, and the commercial sector, which builds housing for profit. Collaborative housing is a form of housing provision based on the idea of a non-profit, participatory and community character of the project and initiated by groups of future residents (Twardoch 2019; Czischke et al. 2016). It can be understood as an umbrella term that encompasses a variety of housing forms with different degrees of collective self-organisation (Czischke et al. 2020; Lang et al. 2020). These forms of collaborative housing include cohousing, eco-villages, living groups as community housing initiatives and collective self-development, collective self-help, non-profit housing, housing cooperatives, as well as Community Land Trust as collective self-provision initiatives (Czischke et al. 2021). Building a taxonomy of collaborative housing forms in Europe was based on an expert panel from Western European countries where collaborative housing forms are well established, and national umbrella organisations are in existence (Czischke et al. 2021). This is also a consequence of the fact that the most widespread studies in this field

published in recent years originate from two regions: Western European countries (Put and Pasteels 2021; Droste 2015; Wankiewicz 2015; Bresson and Denèfle 2015) and the United States (Sanguinetti 2015; Williams 2008; Boyer and Leland 2018; Bourdieu 2006; Jarvis 2015; Stoneman and Battisti 2010; Berggren 2020). Interestingly, in the literature on collaborative housing, there are no studies from Central and Eastern Europe, and thus the case study from Poland makes a valuable extension to the contemporary research.

While examining collaborative housing in Poland, the authors highlighted three main features: sharing spaces, broad participation and community functioning. Vestbro (2000) defined this form as 'housing with more communal spaces or collectively organised facilities than in conventional housing'. Moreover, the importance of creating private spaces alongside shared spaces (Lietaert 2010; Marcus 2000; Fromm 2012; Vestbro 2000), broad participation in organisational, decision-making and financial processes (McCamant and Durrett 2011; Bamford and Lennon 2008; Tummers 2016; Williams 2008), as well as non-hierarchical, consensual forms of group decision-making (Cheung et al. 2014; Espinosa and Walker 2013) are emphasised in the contemporary literature. According to the authors, the link between these features of collaborative housing and the study of the diffusion rate of innovation is missing in contemporary research.

## 2. Materials and Methods

In order to elucidate the limitations to the diffusion of collaborative housing in Poland, a qualitative research design was used. The research methods were selected according to Rogers' framework (Rogers 2003). To gain a detailed understanding of the individual experiences and opinions of key actors of collaborative housing in Poland, the authors decided to gather data with the use of in-depth interviews (McGrath et al. 2019; Voutsina 2018). A meeting held with all the actors related to collaborative housing was not a realistic or practical operation. Thus, the actors were selected. The first step of this selection was to make an inventory of all the actors who are involved in housing collaboratives in Poland. Key groups were identified in this case: one of the leaders who initiated the project 'Nowe Żerniki' in Wrocław and ensured its implementation, a resident who lived in 'Kooperatywa Mieszkaniowa Nowe Żerniki' and was involved in the whole investment process and the housing research community, including researchers, architects, NGOs, finally, there was the leader and a resident of the 'Pomorze' housing project in Gdynia (Table 1). In total, six interviews were conducted, with the duration of individual interviews varying from 1 to 1.5 h. Two leaders were selected from Wroclaw and Gdynia, two residents representing collaborative housing initiatives from Wrocław and Gdynia, a representative of a foundation preparing a new housing cooperative project, as well as a researcher and an architect working on affordable housing.

**Table 1.** Interviews with the actors involved in housing collaboratives in Poland.

| Category | Name | Identity Characteristics | Date |
|---|---|---|---|
| Leader of 'Nowe Żerniki' | Leader 1 | Architectural Bureau in Wrocław | December 2020 |
| Residents | Resident 1 | Resident of 'Kooperatywa Mieszkaniowa Nowe Żerniki' in Wrocław | December 2020 |
| Key informants | Researcher and architect | Silesian University of Technology (Gliwice) | December 2020 |
| | NGO's informant | Habitat for Humanity (Warszawa) | December 2020 |
| | Leader 2 | Leader of 'Pomorze' (Gdynia) | December 2020 |
| | Resident 2 | Resident of 'Pomorze' (Gdynia) | December 2020 |

The design of the in-depth interviews was based on factors that can limit the diffusion of innovation according to Roger's framework, thus affecting the lock-in mechanism. The

factors were used in order to construct the key questions for the in-depth interviews conducted:

(1) Observability feature: In your opinion, are collaborative housing initiatives popular among the members of the social system, such as: neighbours, local community, local authorities, media?

(2) Complexity feature: How do you assess the understanding of collaborative housing? In your opinion, do the members of the social system understand the principles and design of such projects? What elements constitute the biggest problem in understanding this type of undertakings and the lack of willingness to join them?

(3) Compatibility feature: What are the expectations of future residents towards the planned investment? Is the implementation of the project in line with the expectations? Do future residents have an influence on the shape of the given investment, and at what stage?

(4) Trialability feature: Are you guided by other similar collaborative housing initiatives in supporting this type of venture? In your opinion, are pilot programs important in popularising this type of initiative? Do such ventures require institutional support?

(5) Relative advantage feature: Why do you think it is worth choosing this form of housing? How is it better than other forms? What advantages and disadvantages do you see for the operation of a cohousing initiative in times of the SARS-CoV-2 pandemic?

The results of the in-depth interviews reveal the perspective of the key actors involved in collaborative housing. Therefore, their opinions were confronted with individuals who will enter the housing market in the near future and make decisions on how to meet their housing needs. For this purpose, the focus group interview method was used with 42 students of economics, aged 20–21, from the Poznan University of Economics and Business in Poland. The data were collected through a structured group interview process in which several students were interviewed together. With focus group interviews, exchange interaction was achieved in groups, allowing the observation of data less accessible in individual interviews. We took the role of moderators, trying to stimulate a flexible and exploratory discussion that evoked lively interaction between the participants, and not just a dialogue between an interviewer and interviewees (Flick et al. 2004; Linhorst 2002). The respondents had knowledge of real estate markets, investment profitability assessment and corporate social responsibility. They were at the stage of deciding how to meet their housing needs. The survey took place on 11–12 May 2021. In each group of six students, the survey took place for 1.5 h with a moderator (Rabiee 2004). The discussion was based on the results of the in-depth interviews with key informants.

In the final phase of the focus group interviews, each group of students had to create a buyer utility map for collaborative housing in Poland. The students were asked to define the usefulness of this form of housing based on the interviews with the actors involved in housing collaboratives in Poland. They used the buyer utility map created by Kim and Mauborgne (2004, 2017) that was adapted by the authors to a collaborative housing assessment. The buyer utility map consists of two axes: the horizontal axis is the resident experience cycle, and the vertical axis is the utility levers. The resident experience cycle can be divided into six distinct stages: 1. Design: How easy is it to design a dwelling in the collaborative housing scheme? 2. Implementation: How easy is it to carry out the project? 3. Use: How easy is it to live in a dwelling in the collaborative housing scheme? 4. Extras: What other things are required to live in a dwelling in collaborative housing? 5. Maintenance: How easy or difficult is it to maintain a dwelling in collaborative housing? Disposal: How easy is it to dispose of a dwelling in collaborative housing? Utility levers facilitate the discovery of limitations in buyer utility. Productivity: anything associated with efficiency—less time, effort, money—in fulfilling buyers' needs. Simplicity: anything that makes the resident's life easier by eliminating or minimising hassle or complexity. Convenience: this lever focuses on the resident's convenience by saving them frustration and wasted time. Risk reduction: which includes ways to reduce the risks associated

with buying or living in a dwelling in collaborative housing. These risks include financial, physical and reputational aspects. Fun and image: refer to issues such as the look, feel and attitude conveyed by a dwelling in collaborative housing. Environmental friendliness: refers to how 'green' a dwelling in collaborative housing is (Kim and Mauborgne 2017, p. 148). Combined with the resident experience cycle and usability levers, we obtained 36 potential usability spaces. By looking at these usability spaces, the authors have identified the main pain points and utility spaces of collaborative housing.

## 3. Results

### 3.1. Public Awareness of Collaborative Housing

Regarding the results of the in-depth interviews with the key actors of the 'Nowe Zerniki' collaborative housing in Wrocław, it shall be emphasised that public awareness of collaborative housing is low.

Local authorities are not aware of what collaborative housing is and how it works. Moreover, there is also a lack of conviction about the benefits of such solutions functioning in the urban space. Consequently, there are serious difficulties in finding a supportive environment for collaborative housing projects. There is a need to locate this type of housing investment in a friendly environment that supports innovative housing initiatives. In the authors' opinion, collaborative housing in Poland is not a bottom-up approach but a place-based strategy. This means that it requires a friendly environment to be implemented, with a specific strategy for supporting such projects followed by the city. Collaborative housing requires gathering sufficient social capital, people and institutions that will support this type of project. The project leader is not enough for such housing projects to come into being. The respondents indicated that cities/communes were not interested in this type of investment. Following Leader 2's response:

> The deputy mayor of the city was here and she said: "Indeed, an interesting investment, but what will the city gain from it?" The deputy mayor of a large city does not understand that the city, the local government, the authorities have this servant role to the society. And the question is, what will the city gain from it—well, its inhabitants will have apartments.

Cities are not open to innovation, on the one hand, due to the fear of being accused of mismanagement and, on the other hand, due to the lack of instruments, including the abolition of the right of perpetual usufruct of land by the state (Researcher and architect):

> Perpetual usufruct was abolished, which was very helpful and prevented any future financialisation of these units. And the fact that we have terribly little independence of cities, where officials are afraid to resort to such innovations, so as not to be accused of some mismanagement. Someone might challenge this.

(Leader's 1) response completes this matter:

> Our government abolished perpetual usufruct two years ago, and such a preference cannot be applied to collaborative housing anymore. Therefore, on the one hand, various regulations are created to support and encourage such housing, and on the other hand, tools that can realistically help in the formation of collaborative housing are taken away.

The plots for collaborative housing projects in 'Nowe Żerniki' were sold by the city on the basis of a tender that excluded commercial entities. When selecting the best offer, not only was the price taken into account, but also the concept (projections and visualisation), as well as the number and quality of the programmed common areas. The prerequisite for entering the tender was a signed agreement on joint implementation of the project and a financial investment plan (Habitat for Humanity Poland 2021). As stressed by a respondent (Leader 1), even at the time of the sale of plots of land intended for collaborative housing in 'Nowe Żerniki', there were questionable operators who actually intended to carry out developments strictly on profit principles. Of course, such actions were ruled out by the city with many misunderstandings among the 'market' participants in the proceedings.

One of the significant factors related to complexity is the reluctance of commercial banks to grant housing loans for the implementation of projects under collaborative housing. This is mainly due to the lack of detailed regulations in the form of an Act for this type of housing initiative in Poland. Credit institutions are not flexible in financing housing if the projects go beyond the traditional formula of building owner-occupied housing or building rental housing. Any other formula for the implementation of undertakings is rejected or considered in the case when the undertaking is guaranteed by the city. One of the respondents (Resident 1) pointed to the lack of a financial product aimed at collaborative housing:

> When we were looking for funds to make this building, bank clerks were astonished and the answer was: "Sir, there are twelve people and I have only three boxes to enter the names".

The experience of Leader 2 of the 'Pomorze' housing initiative confirmed the fact that due to little knowledge of the idea of collaborative housing, access to financing is difficult:

> When we carried out the first investment, I understood that it is impossible to get such a loan for collaborative housing, as it is not a well-known economic phenomenon or an economic event, and you cannot go to a large network bank, because large network banks, have to prepare a financial product, which is, for example, a mortgage loan for collaborative housing, and this preparation process was done, so to speak, before it enters the cashier, i.e., 3, 4, 5 years earlier. They have their own structures: 100,000 meetings, analyses, lawyers, research somewhere.

As also underlined in (Researcher and architect):

> It seems to me that it is still a matter of the lack of this law, [being the reason why] banks have no basis to grant loans, because they do not know the purpose.

These contributions showed that collaborative housing is completely misunderstood by the environment, and especially by financial institutions. The implementation of innovative housing projects in Poland on general principles is a significant barrier to the development of this type of undertaking. The importance of cooperative banking, which is much more flexible than commercial banking in innovative residential projects, shall be emphasised here. Cooperative banks, due to their specificity, including a limited number of board members, easier access to the president and the possibility of discussing the offer, decide to support housing cooperatives more often. Leader 2's response:

> I then started a conversation with a cooperative bank, because these are small structures. There is a cash desk on the ground floor, and the president is already on the floor above. Of course, it's not that easy to get to him, but the chance of getting to the president of a cooperative bank is a million times greater than [the chance of] getting to, for example, the president of PKO Bank Polski.

Resident 1 also referred to this issue:

> We got financing from a cooperative bank, a small one, in one of the poviat towns near Wrocław, where people were very open and they invited me and a friend to a meeting of the three-person board, during which we talked about the project in detail. They showed great curiosity and decided to finance this investment with a mortgage.

### 3.2. Ability to Use and Adapt Collaborative Housing

Regarding the results of the in-depth interviews in the second area of use and adaptation, collaborative housing rather succeeds in matching the housing project to the expectations of future residents. For one of the four collaborative housing projects in 'Nowe Żerniki', problems arose in the later stages of the housing project. In the authors' opinion, the form of the collaborative housing cannot be assessed 'ex ante'. In the case of 'Nowe Żerniki', three buildings function very close to the cohousing form, and one project has

transformed over time into a form of collective self-development, similar to the 'Pomorze' housing project in Gdynia.

When assessing the compatibility feature, the respondents pointed to the possibility of shaping the space and influencing future housing. Resident 2:

> *It was at the stage of joint planning. My wife is an interior designer and she said that a given colour . . . Well, instead of brown, she said that grey and white would be prettier, more up-to-date. She visualised the building, gates, everything, and everyone liked it, so it was unanimous.*

Resident 2:

> *As this process continued, more children were born, people were promoted at work, etc. These apartments grew a lot during the design work. We started with approximately seventy-square-meter apartments, and ended up on average at one hundred meters. We fulfilled many of our dreams with this design.*

NGO's informant:

> *Common spaces, both inside and outside the building, seem to be tailored to this group. ( . . . ) A collaborative group takes part in the design process. And the produced apartments meet the needs of each member of the group.*

With regard to trialability, it should be noted that the respondents emphasised the cooperation between the future residents, the architectural bureau with its knowledge from other projects and the city. According to Leader 1:

> *Experimentation is not about creating theoretical things on paper, because then they are not subject to any verification; there is simply no feedback from the residents or users. And in order to check such things, to build them, and then maybe to implement them on a larger scale, to show the so-called good practice or an example which can open [people's] eyes, we have to build it.*

*3.3. Cost-Effectiveness of Collaborative Housing*

Regarding the results of the in-depth interviews in the third area of the cost-effectiveness of collaborative housing in relation to other forms of satisfying housing needs, an essential condition is a friendly institutional environment supporting the creation of relationships conducive to housing innovation. Without this environment, a housing project will just boil down to the construction of low-cost dwellings without the developer's margin. A resident of the 'Nowe Żerniki' collaborative housing emphasised that there are no conflicts and that there are even joint initiatives to purchase appliances. Resident 1:

> *We have a large garden where we have a common trampoline, for example. It is an expense of PLN 1500, and if five families chip in, then it's three hundred zlotys each, which is affordable. We have a room on the ground floor, which is called a common room, but it hasn't been finished yet due to lack of funds. We have storage rooms, one on each floor, shared by two families. We don't have any problems with that either—we divide and share the shelves among each other. Nothing has gone missing.*

In other housing projects, there are also no problems with the relations among neighbours. As Resident 2 highlights:

> *Our child calls each of our neighbour's 'aunt' or 'uncle'. If you need to run some errands, and you don't have anyone to leave your child with, someone will always help you, take your child for an hour or two and the child does not feel embarrassed. So, this is such a nice team of people. It is such a small community that it is just perfectly integrated.*

Leader 1 has also confirmed the above assumptions:

> *Certainly, it is also safer, because they also know the circle of friends . . . I noticed that there is absolutely no problem with any thefts, etc., but in this collaborative housing everyone knows one another. Not only that, they know their neighbours' children's friends,*

> *so when someone shows up and is not from this circle, they automatically ask: "What are you doing here?"*

There is a risk of a new resident not adapting to the existing and close-knit housing community. Such a situation took place in one collaborative housing at the 'Nowe Żerniki' estate, where individual apartments for sale appeared Leader 2:

> *There is a simple fact that proves it, namely that there suddenly appeared apartments for sale there, probably two apartments. It is not a lot, but still . . . So, the people who started the collaboration there want to get rid of these apartments now.*

During the SARS-CoV-2 epidemic, the residents of the collaborative housing under analysis greatly appreciated this form of housing Resident 1:

> *Some neighbours were in quarantine, so other people helped them with their shopping. Also, for example, access to the garden, ( . . . ) especially since communication was quite free—you could say, for example, you can go out then and there, we'll go out at other times, so the kids won't see each other there, and in the meantime, it will fade away a bit. This is for quarantine. On the other hand, during the whole spring, when there weren't many cases of the disease, but the fear was very high, ( . . . ) we could get along much better with our friends, when it comes to the rules of meeting [and] spending time together.*

### 3.4. Pain Points and Utility Spaces of the Collaborative Housing

The in-depth interviews conducted served as the preliminary material for discussions with potential residents of such projects. The 42 selected students were divided into seven groups of six people each. The outcome of the discussion in each focus group was a utility map. Based on the resulting utility maps from each group, we made a summary presented in Figure 1. The utility space which the collaborative housing focused on is marked in green, and the pain points that block resident utility is marked in red.

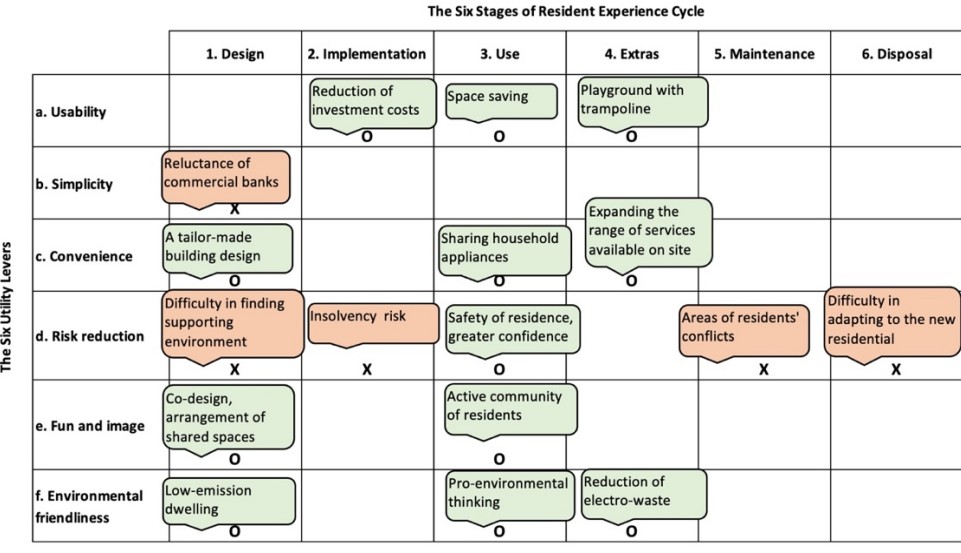

X - pain point that blocks resident utility
O - utility space the collaborative housing focuses on

**Figure 1.** The residents' utility map of collaborative housing in Poland.

Thus, the main utility constraints of collaborative housing have been identified. The greatest pain points were identified at the design stage. Moreover, constraints also exist in the implementation, maintenance and disposal phases. For potential residents, the key pain point emerging in discussions was the difficulty in communicating with neighbours. The respondents were terrified of having to resolve possible conflicts. The issue of resolving difficult situations in daily life came up in the discussions. This issue occurred again in

the implementation of the project. The respondents highlighted the risk of insolvency of collaborative housing project participants in the implementation phase. Again, the issue of lack of trust in project participants and the fear of their financial difficulties came up in the initial phases.

The key utilities on which the 'Nowe Żerniki' collaborative housing is based on, are found primarily in the process of designing and living in dwellings. First of all, the residential space and common areas are tailored to the needs and expectations of future residents while maintaining low emissivity of the buildings. The possibility of saving space and sharing household appliances is of key importance. The relevance of shared pro-ecological values is underlined. The discussion focused very strongly on this thread. The strong connection between housing innovation and ecology (low emission, pro-environmental thinking, reduction of electro-waste) is the most significant utility for the respondents. Interestingly, a playground with trampolines plays a crucial role for potential residents (this aspect was dominant in the utility maps). It is due to the fact that playgrounds are usually the most common shared spaces in Poland.

## 4. Discussion

In the literature, collaborative housing is the result of bottom-up initiatives (McCamant and Durrett 2011). We have proved that the development of collaborative housing in Poland requires a place-based approach instead of a bottom-up approach. According to (Barca 2009, p. 4), the place-based policy is 'a long-term strategy aimed at tackling persistent underutilisation of potential and reducing persistent social exclusion in specific places through external interventions and multilevel governance'. This policy is based on the locality enabling the development of its strategic potential with the use of its territorial capital and local knowledge (Weck et al. 2021; Piras et al. 2021; Borén and Schmitt 2021). The connection to the place (Wrocław as the city and 'Nowe Żerniki' as a district) and the local capital (especially innovative architectural ideas) were crucial factors for all the key actors of the collaborative housing process in 'Nowe Żerniki'.

Having conducted the empirical research, the authors recognise the potential and interest of collaborative housing in Poland, with the condition for implementing this type of investment being the creation of a friendly legal and institutional environment that will support future residents in the process of creating the investment. Without instruments aimed directly at collaborative housing, buildings will be made in the quasi-collaborative formula, resembling commercial construction in an individual dimension, not meeting the criteria that constitute collaborative housing.

Collaborative housing is often lauded as an alternative with the potential to develop more socially and ecologically sustainable neighbourhoods (Giorgi 2020). The uncontrolled development of technology, climate change, the complexity of urban life, the crisis of traditional communities and the inability to create lasting, appropriate human bonds seem to be greater concerns for collaborative housing in more developed countries (Bresson and Denèfle 2015; Bresson and Labit 2020; Chiodelli and Baglione 2014). Residents need to take part in common work, maintenance of buildings and outdoor areas and participate in the housing associations' democratic decision-making (Sørvoll and Bengtsson 2020). What appears to be the greatest asset of collaborative housing residents in more established housing systems poses a problem for residents in countries with low social capital and traditional attitudes to home ownership.

The future of cities depends on how well and how soon access to adequate housing will be provided for everyone. Positioning housing at the centre of national and local urban agendas will be instrumental for achieving this goal and promoting inclusion and equality of opportunities in the urban development process (United Nations Human Settlements Programme 2015). Collaborative housing could be one of many solutions in this regard.

In our study, in-depth interviews were conducted only with the representatives of collaborative housing communities. It may be a limitation in transferring the findings to the entire population of residents of collaborative housing in Poland.

For the research based on focus groups, students from the third-best school of economics in Poland (according to the Educational Foundation Perspektywy, 2021), aged 20–21, were selected. Three student practice groups were sampled, with a total of 42 students. The study was conducted in small teams of six. However, the method of selection may create a limitation in transferring the findings to the entire population of young people in Poland.

**Author Contributions:** Conceptualization, P.L.; methodology, P.L. and Z.R.; software, P.L., Z.R. and K.S.; validation, P.L., Z.R. and K.S.; formal analysis, P.L. and Z.R.; investigation, P.L. and Z.R.; resources, P.L., Z.R. and K.S.; data curation, P.L., Z.R. and K.S.; writing—original draft preparation, P.L.; writing—review and editing, P.L., Z.R. and K.S.; visualization, P.L., Z.R. and K.S.; supervision, P.L., Z.R. and K.S.; project administration, Z.R.; funding acquisition, P.L. All authors have read and agreed to the published version of the manuscript.

**Funding:** The author acknowledges financial support within the Regional Initiative for Excellence programme of the Minister of Science and Higher Education of Poland, years 2019–2022, grant no. 004/RID/2018/19, financing 3,000,000 PLN.

**Institutional Review Board Statement:** The article meets the requirements of the Declaration of Helsinki and complies with the regulations of the Ethics Committee of the Poznan University of Economics, Rector's Order No. 4/2017.

**Informed Consent Statement:** Informed consent was obtained from all subjects involved in the study.

**Data Availability Statement:** https://doi.org/10.5281/zenodo.6244297, accessed on 23 January 2022.

**Conflicts of Interest:** The authors declare no conflict of interest.

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
