# Peer review of "Implementation Risk Factors of Collaborative Housing in Poland: The Case of ‘Nowe Żerniki’ in Wrocław"

_jrfm, doi:10.3390/jrfm15030101_

Round 1

Reviewer 1 Report

What is the significant contribution of the study to the extant body of knowledge on affordable housing?

Actually, the concept of "collaborative housing" is not clear.  Should this type of housing be classified as a kind of "third-sector housing"?  Is is a kind of self-help housing?  Is it something related to public-private partnership?  The authors need to elaborate more the idea of "collaborative housing" in the article.  Also, the authors should link the concept to the wider literature in the field of housing studies.

The referencing is poorly done.  References are missing for a lot of figures and statistics quoted in the manuscript (e.g. in the first few lines of the introduction).  The authors should check the whole manuscript carefully to see if adequate references or sources have been provided.

The research design has not been clearly explained.  In Line 107, the authors said there were "three key groups" of participants.   Who were they?  I could just see the six participants in the paragraph.

How were the six participants selected?  Were they representative of the population?

How were the in-depth interviews and focus group interviews integrated as an overall research strategy?

Why were only students in one university chosen for the focus group interviews?   Why only economics study were eligible for the participation in the research?  How were the students sampled?

What does "environment's understanding" in Line 120 mean?

What does "... to use a dwelling ... " in Line 156 mean?

How could the respondents (students studying economics) know the disposal issue of the collaborative housing?  They have not purchased one.

The paper lacks a clear conceptual framework.

The findings of the research have not be adequately discussed.

What are the limitations of the research?  How do these limitations affect the interpretation and generalization of the research findings?

Reviewer 2 Report

The article contains information technical and innovative. The problem addressed is current and has technical relevance, which makes it significant. The abstract is written concisely. The paper is well organized and convincing. The experimental methodology is described comprehensively. Interpretations and conclusions are justified by the results. 

My recommendations are:

The abstract can be rewritten to be more meaningful should clarify what is exactly proposed (the technical contribution) and how the proposed approach is validated.

Literature review techniques have to be strengthened by including the issues in the current system and how the author proposes to overcome the same.

In the references in the Introduction section, the authors cite some works. However, they have not indicated the advantage or disadvantage and their relations to this paper. It's a little confusing.

Quality of Figures is so important too. Please provide some high-resolution figures. The comparison of different methods using clear graphs should be explained.

The paper does not provide significant experimental details needed to correctly assess its contribution: What is the validation procedure used?

The conclusions are poorly written. The authors need to rewrite the present section and show how the work advances the field from the current state of knowledge.

Discuss the future works concerning the research state of progress and its limitations.

There are some typos and mistakes in the manuscript. Read the manuscript and correct them. 

Reviewer 3 Report

Comments

Over the centuries, housing needs have been sustained in a very limited set of spatial configurations, an adaptive behaviour to each moment of reality, but which has never responded (and will not respond) to people's needs. Added to this, the general shortage of decent and affordable housing, the increase in people living alone and the aging of the population have forced a reassessment of the main models of human habitation. Persistent proposals for solutions that can better respond to the human housing reality come up against large cities. With this in mind, the focus of research has focused too much on contemporary models of collective housing that seek to reinterpret the concept of housing to better adapt to current lifestyles.
We can build housing that works as a community, but we accept, and rightly, that the best houses are those where people live. Another thing is whether they are adaptable to the size of the households.
The genesis of this collaborative housing concept is based on the fact that each family has guaranteed its personal space, with a set of support services, and with defined rules, in which all tasks and activities are shared.
It is an intergenerational facility that includes, e.g., day care centers and spaces for young people on the ground floor, assisted living facilities on the upper floors for the elderly with autonomy or for digital nomads. The doubt is in the immense fringe of people who do not want this to be the preferred involvement, since it is prone to breaches in their privacy.
Each of the apartments is different from the others, as each of the residents can bring their own furniture and personal belongings and decorate to their liking. Afterwards, they share common spaces, share the tasks and organize their parties, where they receive young people and children and their families. All this is very theoretical and difficult to implement in medium and large urban spaces, but it has the virtue of providing a roof for many people spread out spatially in small communities.

The interest in collaborative housing exists and its study has been explored as a recent scientific concept that it is. This concept can attract renovations in cities, especially in historic or very central centers. The intention will be to attract other people, but also to let people stay, creating the conditions for such a desideratum. In these places in the cities, the owners have their buildings that no longer respond to their needs.
I must mention that despite the fact that the scale development of this type of housing is still very tenuous, it is always necessary to look ahead to new housing responses that allow the institutionalization of people to be postponed.
The universe of cooperatives is one of the sectors with great appetite for cohousing. Cooperatives will certainly be available to participate in housing innovation, regardless of the construction model, neighborhood and outdoor spaces.
However, the common criticism is the difficulty in getting this type of projects approved by the municipalities, due to the specificities of this type of equipment. Starting, from the outset, with the difficulty of getting a project approved in which it is not possible to replace the parking space, mandatory by law, using this area for other equipment. This restriction is enough to make the projects unfeasible.

As the authors state in the article, collaborative housing has its own specificities and follows new building models and, therefore, it is necessary to change Building Regulations to include these models of common living. The obstacles in the case of ‘Nowe Å»erniki in WrocÅ‚aw is an example of this.

Suggestions

Housing production has been fostered by public tenders and very little through the market, even if construction and sale are supported by public authorities. One of the innovations is, e.g., in the creation and dissemination of architectural possibilities, which can be used in new prospects, and also through their dissemination, provided that within the scope of a regulation that serves as a basis for other ideas and projects. It is a kind of multiplier effect of innovation in the housing phenomenon.
From the perspective of technical assistance as a public housing policy, local decision-makers can and should coordinate and promote a collaborative environment in which actors are included, e.g. teachers, based on a participatory methodology. They can even propose policies that address the quantitative and qualitative housing deficit through the improvement of precarious housing. Strategies such as technical assistance in this context are a fundamental step in promoting the quality of life of inhabitants who are already established in a certain place, without the need to move them away from urban centers and maintaining the possibility of low-cost housing.
The structure of the interviews was sufficient for the study. In situations that arise, the complexity of the questions and the actors will be different but useful for future studies, especially for those who enter the housing market for the first time or have housing needs, not forgetting the proximity of housing to the workplace, as well as commercial activities and leisure spaces.

In addition to the set of observations and suggestions mentioned above, I present another that consists of two readings: (1) United Nations (2015), Housing at the center of the New Urban Agenda, United Nations Human Settlements Programme, https://unhabitat .org/housing-at-the-centre-of-the-new-urban-agenda and (2) Czischke, D., Zijlstra, S., & Carriou, C. (2016), The rise of collaborative housing approaches in England, France and the Netherlands: (How) are national housing policies responding? Paper presented at the workshop on Collaborative Housing, ENHR conference “Governance, Territory and Housing”, Belfast (Northern Ireland), 28 June – 1 July 2016.

Round 2

Reviewer 1 Report

Although the authors attempted to address my concerns in the resubmission, I still think they failed to clarify the irregularities or ambiguities.  For example,  

1) The contribution of the study to the extant body of knowledge on affordable housing is still unclear.  I cannot see how the current study can be tied to the existing literature on "collaborative housing".

2) Still, the authors should have followed the referencing style set out for the journal strictly. 

3) The conceptual framework is still not clear enough. The discussion of the research findings should correspond to the conceptual framework.

4) While the authors named some of the minor research limitations, they avoided mentioning some more critical limitations.  Also, they did not discuss how these limitations affect the interpretation and generalization of the research findings.

Reviewer 2 Report

The authors made the various requested changes. I am in favor of the publication.

Author Response

Thank you for your revision and approval our paper.

Round 3

Reviewer 1 Report

I don't have any further comments on the paper.